# POSITION: MULTI-FACETED STUDIES ON DATA POISONING CAN ADVANCE LLM DEVELOPMENT

## ABSTRACT

The lifecycle of large language models (LLMs) is far more complex than that of traditional machine learning models, involving multiple training stages, diverse data sources, and varied inference methods. While prior research on data poisoning attacks has primarily focused on the safety vulnerabilities of LLMs, these attacks face significant challenges in practice. Secure data collection, rigorous data cleaning, and the multistage nature of LLM training make it difficult to inject poisoned data or reliably influence LLM behavior as intended. Given these challenges, this position paper proposes rethinking the role of data poisoning and argues that **multi-faceted studies on data poisoning can advance LLM development**. From a threat perspective, practical strategies for data poisoning attacks can help evaluate and address real safety risks to LLMs. From a trustworthiness perspective, data poisoning can be leveraged to build more robust LLMs by uncovering and mitigating hidden biases, harmful outputs, and hallucinations. Moreover, from a mechanism perspective, data poisoning can provide valuable insights into LLMs, particularly the interplay between data and model behavior, driving a deeper understanding of their underlying mechanisms.

## 1 INTRODUCTION

Data poisoning Zhao et al. (2023b); Zhang et al. (2023); Kojima et al. (2022), which refers to the threat model that introduces maliciously crafted data into model training processes Zhao et al. (2024b); Kandpal et al. (2023); Hubinger et al. (2024), has brought great threats to the security and trustworthiness of LLM applications. Recent studies have shown that such poisoned data can have far-reaching consequences in LLMs, including performance degradation (He et al., 2024d), the insert of backdoors that allow attackers to control outputs under specific conditions (Wan et al., 2023; Kandpal et al., 2023; Xiang et al., 2024), and the manipulation of responses to serve malicious purposes (Bekbayev et al., 2023; Rando & Tramèr, 2023; Bowen et al., 2024a).

Unlike conventional machine learning models, LLM development usually undergoes a much more complex lifecycle. This includes pre-training on large-scale datasets, instruction tuning and RLHF Ziegler et al. (2019); Ouyang et al. (2022), fine-tuning for specific tasks or domains (Hu et al., 2021; Liu et al., 2022), inference-time adaptation methods such as in-context learning (ICL) (Brown et al., 2020), and applications such as retrieval-augmented generation (RAG) (Lewis et al., 2020) and LLM agents (Wu et al., 2023; Gao et al., 2024). Since diverse data is involved in multiple stages of LLM's lifecycle, data poisoning attacks naturally extend from attacking one dataset to all data sources in the lifecycle, and we refer to this extended attack as **lifecycle-aware data poisoning for LLMs** (detailed in Section 2). This broader scope introduces new aspects for investigation.

However, the majority of existing data poisoning research on LLMs holds a threat-centric perspective that focuses on uncovering the risk of data poisoning, and mainly adopts attacks designed for traditional machine learning models to LLMs. We identify two fundamental limitations of the existing threat-centric efforts as follows:

First, an often unjustified assumption is that attackers can directly or indirectly manipulate data. This assumption is especially challenging for LLMs, as their data sources are highly diverse and often private. For instance, large organizations developing LLMs typically do not disclose their pre-training or post-training datasets. This applies to both open-source models, such as the Llama series (Dubey et al., 2024), and API-only models, such as GPTs (Achiam et al., 2023) (more details in Section

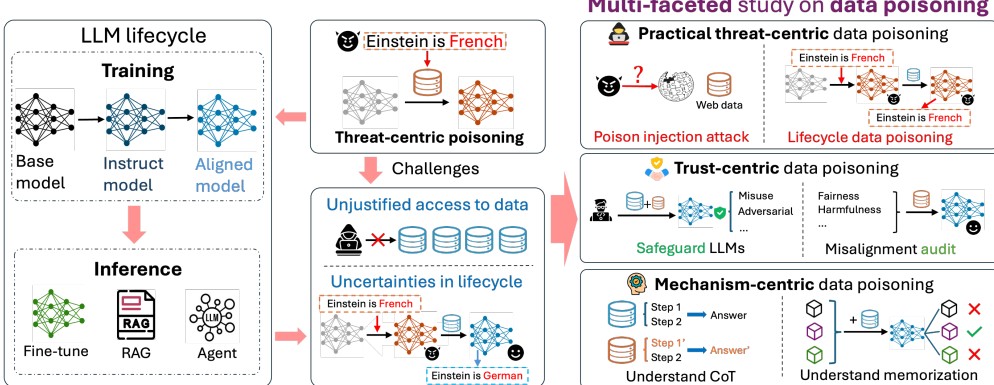

Figure 1: An illustration of this paper's structure. (Left) LLM's lifecycle including multiple training and inference stages (Section 2.1). (Middle) Threat-centric data poisoning and its challenges (Section 2.2). (Right) The **multi-faceted study on data poisoning**, including practical threat-centric (Section 3), trust-centric (Section 4) and mechanism-centric data poisoning (Section 5).

2). If it is not well-justified whether the attacker is able to manipulate the data, the feasibility and impact of data poisoning attacks in real-world scenarios cannot be properly estimated, potentially overlooking the scenarios that are more likely to happen. Second, the multiple stages of an LLM's development lifecycle introduce significant uncertainties, such as variations in training algorithms in different stages. Since attackers usually lose control over poisoned datasets once they are integrated into complex training pipelines, these uncertainties will undermine the effectiveness of data poisoning attacks throughout the later stages. Specifically, compared to traditional machine learning models, which often follow a training-and-testing paradigm that better preserves poisoning effects (He et al., 2023), the complicated processes within LLMs make it difficult for attackers to account for all factors. For example, poisoned data injected during the instruction tuning stage may be overwritten by diverse datasets and alignment objectives in the preference learning stage (Wan et al., 2023). Furthermore, unknown downstream tasks and datasets during inference-time adaptations can further dilute poisoned patterns (Qiang et al., 2024).

These limitations motivate us to rethink data poisoning in the era of LLMs by investigating two critical questions. First, the lack of proper justification of the attacker's capability to directly manipulate data and the challenge of sustaining the poisoning effect across LLMs' lifecycle inspires: *(Q1) How can we enhance the practicality of data poisoning attacks to position them as a real-world threat?* This question inspires us to explore practical threat models and effective strategies to reveal data poisoning risks in real-world scenarios. Second, despite the practical challenges for attackers, existing research also fails to fully leverage insights into LLM vulnerabilities from data poisoning to address broader objectives, such as developing trustworthy LLMs. Therefore, we aim to investigate: *(Q2) Can data poisoning serve as a tool to advance LLM research beyond conventional threat-centric perspective?* This question changes the focus from threats to opportunities, focusing on how data poisoning can be leveraged to guide trustworthy LLM development, and even understand LLM mechanisms.

To address (**Q1**), we advocate for developing realistic strategies, such as the proposed *poison injection attack* (detailed in Section 3). Practical strategies should go beyond focusing solely on the consequences of poisoning. They need to consider LLM-specific development scenarios and security measures to enable effective data injection. Additionally, these strategies aim to sustain poisoning effects throughout the LLM development lifecycle. By targeting vulnerabilities such as web crawling pipelines (Carlini et al., 2024) and agent memory storage systems (Chen et al., 2024b), which are essential parts of LLM data collection, these strategies validate the feasibility of data poisoning attacks, transforming theoretical threats into real-world risks.

For (**Q2**), we reconsider key characteristics of data poisoning attacks, including the ability to exploit model mechanisms (Steinhardt et al., 2017; Yu et al., 2022; He et al., 2024d), dependence on strategic data selection (He et al., 2024b; Xia et al., 2022; Zhu et al., 2023), and capacity to precisely control model output (Schwarzschild et al., 2021; Shafahi et al., 2018; Geiping et al., 2020). Specifically, we propose leveraging data poisoning techniques to advance LLM trustworthiness and recognize it as a powerful lens for understanding model behavior. We refer to these novel perspectives as **trust-centric**

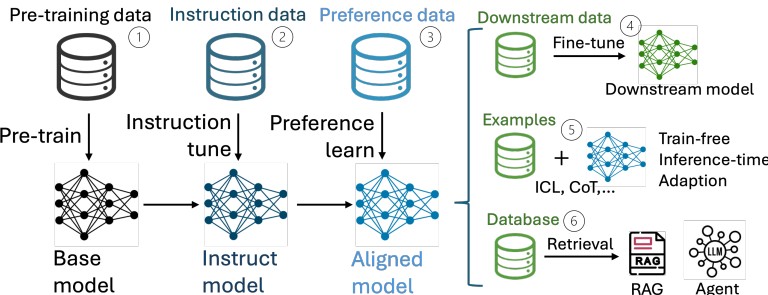

Figure 2: A systematic overview of an LLM's development lifecycle including training stages (pre-training, instruction tuning, preference learning) and various inference stages such as fine-tuning, train-free inference-time adaption and retrieval-based applications (show inside the right brace).

(Section 4) and **mechanism-centric** (Section 5) respectively, to distinguish them from the traditional threat-centric view.

Trust-centric data poisoning leverages data poisoning techniques to address security threats and misaligned behaviors like fairness (Li et al., 2023), misinformation (Chen & Shu, 2023) and hallucination (Yao et al., 2023) in LLM outputs. This can be achieved by embedding specially designed data into clean datasets to influence model behavior. For example, secret tasks (China Daily, 2024) can be injected during LLM training to protect proprietary models. Similarly, backdoored models can mitigate jailbreak attempts by triggering predefined safety responses to malicious prompts (Chen et al., 2024a; Bowen et al., 2024a). Beyond security, trust-centric data poisoning can address biases in training data and eliminate misaligned patterns (Zhang et al., 2024a) by injecting corrective data.

Mechanism-centric data poisoning focuses on understanding LLM behaviors, such as Chain-of-Thought (CoT) reasoning (Wei et al., 2022) and long-context learning (Li et al., 2024b). Its key advantage is precise control over data manipulation, allowing the creation of "poisoned datasets" to study how specific data patterns influence model behavior. For instance, to examine which reasoning steps are critical or whether incorrect examples aid reasoning, we can perturb individual steps in few-shot examples and test model sensitivity (Cui et al., 2024; He et al., 2024a). This controlled approach enables fair comparisons of each step's influence on CoT reasoning. Additionally, this perspective sheds light on LLM memorization by injecting patterns into training data and evaluating their effects, offering insights into how LLMs encode and retrieve information from training samples.

In summary, these discussions argue that **multi-faceted studies on data poisoning can advance LLM development**. As shown in Figure 1, the rest of the paper is organized as follows. In Section 2, we provide a holistic overview of data poisoning attacks on LLMs, and discuss fundamental limitations. In Section 3, we discuss practical threat-centric data poisoning. In Section 4 and 5, we introduce two novel perspectives: trust-centric data poisoning and mechanism-centric data poisoning that extend data poisoning methods from threats to useful tools that develop more trustworthy LLMs and help understand LLMs.

## 2 DATA POISONING IN LLMS

In this section, we present a comprehensive overview of data poisoning in LLMs, organized by stages of an LLM's lifecycle. Following this, we discuss the limitations of existing studies of data poisoning.

### 2.1 AN OVERVIEW OF DATA POISONING IN LLM'S LIFECYCLE

Generally speaking, data poisoning attacks aim to inject maliciously designed data (known as poisoning data) into the training set to achieve the attacker's malicious goals. These goals often range from degrading the model's performance (targeted and untargeted attacks)(Shafahi et al., 2018; Fowl et al., 2021) to triggering specific behaviors (backdoor attacks)(Schwarzschild et al., 2021; Gu et al., 2019). Since LLMs are commonly pre-trained on large-scale datasets that are scraped from the Internet and can be contaminated by attacks (Carlini et al., 2024), data poisoning attacks have also captured increasing attention in the era of LLMs (Wan et al., 2023; He et al., 2024d).

Table 1: A summarization of threat models in existing threat-centric data poisoning for LLMs. We focus on attackers' capability on data and models, where Partial access represents scenarios that attackers can inject a proportion of poisoned samples or modify a subset of clean data. Full access means complete control over data and LLMs.

| Data access | Model access | LLM lifecycle Stage | References |
|---|---|---|---|
| Partial access | No access | Pre-training | (Zhang et al., 2024b; Hubinger et al., 2024) |
| | | Instruction tuning | (Wan et al., 2023; Xu et al., 2023; Shu et al., 2023; Qiang et al., 2024; Yan et al., 2024) |
| | | Preference learning | (Wu et al., 2024; Rando & Tramèr, 2023; Baumgärtner et al., 2024) |
| | | Inference (fine-tuning) | (Zhao et al., 2024a; 2023a; Bowen et al., 2024a) |
| | | Inference (ICL, CoT) | (He et al., 2024c; Xiang et al., 2024) |
| | | Inference (RAG) | (Zou et al., 2024; Xue et al., 2024; Chen et al., 2024c) |
| | | Inference (Agent) | (Chen et al., 2024b) |
| Full access | No access | Inference (fine-tuning) | (Halawi et al., 2024; Huang et al., 2024b; Bowen et al., 2024a) |
| Full access | Full access | Preference tuning | (Shi et al., 2023; Wang & Shu, 2024) |
| | | Inference (fine-tuning) | (Kandpal et al., 2023; Bowen et al., 2024a; Li et al., 2024c; Liu et al., 2024a) |
| | | Inference (Agent) | (Wang et al., 2024; Yang et al.) |

Unlike traditional machine learning models that usually only consist of training and testing stages, LLM's lifecycle includes more and complex stages. As shown in Figure 2, stages in an LLM's lifecycle include different training stages: (1) pre-training stage where a base model is trained on large-scale pre-training datasets from scratch via next-token prediction; (2) instruction tuning stage where the base model is fine-tuned on the instruction data to obtain the instruction-following capability; (3) preference learning stage where the instruct model is tuned to align with the human preference on the preference data which are human annotated. There are also various kinds of inference stages: (4) downstream fine-tuning that finetunes the LLM on downstream datasets for a specific downstream task; (5) train-free inference-time adaptions such as ICL or CoT where examples are used to adapt tasks without changing model parameters; (6) retrieval-based applications such as Retrieval-augmented generation (RAG) and LLM agents which retrieve from external databases to help execute tasks. Existing literature reveals the harmful impact of injecting poison into the data in these stages, e.g., (Wan et al., 2023; Kandpal et al., 2023; Hubinger et al., 2024; Zou et al., 2024). Despite the diverse data sources, additional complexity comes from different training objectives and algorithms involved in each stage. For instance, pre-training is conducted on large-scale unlabeled data via next-token prediction; instruction tuning and preference learning rely on annotated data and supervised algorithms like Supervised Fine-Tuning (SFT) (Touvron et al., 2023) and Direct Preference Optimization (DPO) (Rafailov et al., 2024).

The diverse data sources and training objectives of LLMs make them highly susceptible to a broader range of data poisoning attacks, collectively termed as **lifecycle-aware data poisoning for LLMs**. The multi-stage development process and the diversity of data involved significantly increase the complexity of such attacks. Our investigation reveals that most existing studies (Yao et al., 2024; Das et al., 2025; Chowdhury et al., 2024; Zhang et al., 2025; Zhao et al.) on data poisoning in LLMs adopt a **threat-centric** perspective which treats data poisoning as an adversarial act. These approaches often rely on traditional data poisoning methods (Das et al., 2025; Zhao et al.) without adequately addressing the unique complexities inherent to LLMs as introduced above. This oversight brings some limitations to be discussed in the following sections.

## 2.2 LIMITATION IN EXISTING THREAT-CENTRIC DATA POISONING

Lifecycle-aware data poisoning for LLMs is far more complex, yet most existing approaches still rely on threat models and methods designed for traditional attacks. We identify two key limitations in this approach: (1) insufficient justification for the practicality of the threat models; and (2) the challenges posed by amplified uncertainties across the multiple stages of LLMs.

### 2.2.1 ANALYZING THE PRACTICALITY OF DATA POISONING THREAT MODELS

Data poisoning attacks involve manipulating data, either by directly modifying existing datasets or injecting malicious data. This raises a critical question about threat-centric research: *Are the assumptions about an attacker's access to data practical?* To answer this question, we summarize threat models in existing works, as shown in Table 1.

According to Table 1, most threat models presume that the adversary can directly/indirectly inject or modify the clean data. This assumption has been widely adopted by poisoning attacks in all stages of the LLM's lifecycle. In practice, data is often regarded as a highly valuable resource. Unlike the

assumptions commonly made in data poisoning literature, it is typically inaccessible to regular users due to developers' legal and safety concerns. Take the Llama series (Touvron et al., 2023; Dubey et al., 2024) as an example. While much of the pre-training data is mostly crawled from the web, the data undergoes a thorough cleaning process before being used for training (Dubey et al., 2024). This process includes safety filtering to remove unsafe content, text cleaning to extract high-quality data, and both heuristic and model-based quality filtering to eliminate low-quality documents. Post-training data, such as instruction-tuning datasets and preference data, is generated and annotated under the supervision of developers and is also subjected to careful cleaning and quality control. These show that LLM training data is typically under the careful control of model developers, which poses significant challenges to the assumption that attackers can access these training data.

The challenge of the adversary's access to the data is not limited to the training stages, but also the inference stages or downstream adaptions including downstream fine-tuning, ICL and applications like RAG. Data used for downstream fine-tuning, or inference-time adaption like ICL is usually collected by users themselves, and the small size of data[1] (Min et al., 2022) allows for better quality and safety control. The database in the RAG system is also an internal resource (Li et al., 2024a), especially in privacy-intensive domains such as healthcare, education, and finance. Various security measures, e.g., role-based access control (Sandhu, 1998; Ant, 2025) and data encryption (Ramachandra et al., 2022), can prevent adversarial access to the data.

Therefore, we can conclude that the practicality of the assumption allowing attackers to directly/indirectly manipulate data is not properly and sufficiently justified. While some works provide examples to illustrate that this assumption holds under rare scenarios (Chen et al., 2024b; Xiang et al., 2024), more evidence on how data manipulation can be achieved would be helpful in addressing the real concerns of data poisoning.

### 2.2.2 LIMITATIONS DUE TO THE COMPLEXITY OF LLM LIFECYCLE

The complexity of the LLM lifecycle makes it significantly harder for attackers to control the impact of poisoned data. In typical data poisoning scenarios, attackers are assumed to control the data at one stage but lack knowledge of subsequent stages, including the data and algorithms used after the poisoned data is released by the attacker. This assumption is common in traditional data poisoning attacks. Some existing works (He et al., 2023; Huang et al., 2020) focus on developing effective attacks to address uncertainties in traditional models which typically involve only a single training and testing stage. However, the complexity of LLM's multi-stage nature exacerbates this challenge. For example, the pre-training stage mostly leverages unlabeled data for next-token prediction, while the preference learning stage utilizes RLHF or DPO on human-annotated preference data. This complexity makes it far more difficult to ensure that poisoning effects persist across stages, especially when the attacker targeting an early stage has no control over later stages.

To set an example, poisoned data injected during instruction tuning may lose its impact during the subsequent preference learning stage (Wan et al., 2023; Qiang et al., 2024). After this stage, alignment procedures such as RLHF are designed to optimize the model's outputs to align with human preferences, which can effectively dilute or neutralize malicious effects introduced earlier. Consequently, the threat posed by poisoning during instruction tuning is significantly diminished by the time the aligned model is released.

Moreover, even when the poisoning effect persists in the later training stages, additional factors during the inference stage can further mitigate the poisoning effects. For instance, inference methods such as training-free adaptations (e.g., ICL) have been shown in existing works (Qiang et al., 2024) to defend against poisoning attacks injected at the instruction tuning stage. These compounded uncertainties—arising from diverse stages, algorithms, and inference methods—pose significant challenges for attackers attempting to sustain the impact of their poisoning efforts throughout the LLM lifecycle.

## 3 PRACTICAL THREAT-CENTRIC DATA POISONING

Due to the aforementioned limitations, it is desired to explore more practical data poisoning for LLMs, **practical threat-centric data poisoning**. It aims to investigate data poisoning threats in realistic scenarios. Next, we demonstrate our concept with the following two aspects.

---

[1]Existing works have illustrated that a few examples are sufficient for ICL and CoT.

**Poison injection against secure data collection**   A key interest of practical threat-centric data poisoning is its emphasis on validating both the feasibility and practicality of attacks. It advocates for practical *poison injection attacks*, which aim to strategically insert malicious data into clean datasets involved in the LLM lifecycle. A successful poison injection attack demonstrates that the victim dataset can be poisoned. To conduct a successful poison injection attack, we suggest identifying and exploiting potential vulnerabilities in data collection, curation, and storage pipelines across the entire LLM lifecycle. We present some illustrative examples from different stages.

- Pre-training: During the pre-training stage, (Carlini et al., 2024) explore strategies for injecting poisoned samples into web-scale datasets by exploiting vulnerabilities in data collection processes. Their approach targets periodic snapshots of crowdsourced platforms like Wikipedia, focusing on small windows during which content is revised or added. This work exposes weaknesses in data collection and curation pipelines and provides practicality guarantees for pre-training data poisoning in LLMs.
- Preference learning: In the preference learning stage, attackers can identify vulnerabilities in the human annotation process for preference data to inject malicious data. This injection can involve exploiting crowdsourcing platforms (such as Amazon Mechanical Turk (Turk, 2012)), infiltrating the annotation workforce by posing as annotators to mislabel texts or introducing ambiguous and highly subjective content for labeling to create systematic biases.
- Train-free inference-time adaptions: In retrieval-based applications, such as LLM agents, attackers can inject poisoned samples during the inference stage solely through user queries. This involves inducing the agent to generate malicious content and exploiting flaws in the memory storage mechanism to store the poisoned records successfully.

**Weaker attacker's ability and new attacking objectives**   Another critical aspect of practical threat-centric data poisoning is the consideration of uncertainties across LLM's life cycle. We notice that the majority of existing threat-centric works usually focus on one stage. In other words, they often assume that the attackers inject malicious samples into the data of one stage and evaluate how poisoned data influence the model behavior after this particular stage (Wan et al., 2023; Kandpal et al., 2023; He et al., 2024d). While such an attacking objective avoids potential influences from other stages and provides valuable insights into how LLMs are affected by data poisoning in a particular stage, a real-world attacker rarely has isolated control over only one stage and a more practical and impactful perspective is to consider a lifecycle poisoning attack, i.e. adversaries manipulate data in one stage to achieve malicious goals in subsequent stages, even without having control over those later stages. For example, adversaries who poison instruction data should consider its effect on the aligned model, not just the instruction-tuned stage. Moreover, inference-stage uncertainties, such as fine-tuning on clean downstream data neutralizing poisoning effects or the resistance of ICL to instruction-data poisoning (Qiang et al., 2024), must also be considered, as discussed in Section 2.1.

Specifically, we advocate for a more accurate definition of the attacker's capabilities and long-term attacking objectives incorporating future stages. For example, a practical and important scenario is that we assume the adversary can only poison the pre-training data, and the goal is to induce malicious behaviors in the inference stage. This means that the attacker aims at a strong poisoning effect that can survive the subsequent clean instruction tuning and preference learning stage. Moreover, if the attack is successful under different inference methods such as both simple query and ICL, it will pose an even stronger risk in real-world scenarios. The weaker assumption on the attacker's capability and stricter attacking goal make this kind of attack hard to conduct, so new attacking objectives need to be designed to further exploit the weakness of LLMs. Inspirations can be drawn from traditional data poisoning attacks like (He et al., 2023; Huang et al., 2020) where uncertainties of algorithms and data are explicitly incorporated in the attacking algorithm.

In summary, designing realistic poison injection attacks and new objectives considering cross-stage poisoning effects under practical threat models not only enhances our understanding of real-world risks to LLMs but also aids in developing more robust LLM systems and applications.

## 4   TRUST-CENTRIC DATA POISONING

In this section, we explore the use of data poisoning to enhance the trustworthiness of LLMs, a novel perspective we term **trust-centric data poisoning**. This perspective aims at utilizing techniques of data poisoning in building robust LLMs, identifying and mitigating potential issues including hidden biases, harmful outputs, hallucinations etc.

Given the different goals of threat-centric data poisoning, the settings for trust-centric approaches are adjusted accordingly. First, the role of the "attacker" in trust-centric data poisoning is broader, encompassing model developers or researchers who have greater control over the data and various stages of the LLM lifecycle. Second, trust-centric data poisoning modifies objectives, such as loss functions, shifting from maximizing the poisoning effect in threat-centric approaches to maximizing resistance to threats and minimizing the occurrence of misaligned behaviors.

Trust-centric data poisoning differs from practical threat-centric data poisoning in Section 3. First, while both trust-centric and practical threat-centric data poisoning are related to the trustworthiness of the model, their paradigms are different: For practice threat-centric data poisoning, one needs to first reveal the vulnerabilities through attacks and then enhance the model robustness correspondingly. In contrast, for trust-centric data poisoning, we directly utilize data poisoning techniques and objectives to improve LLM's trustworthiness, which does not include any attack phase. Second, trust-centric data poisoning considers a broader scope of trustworthiness. In threat-centric data poisoning, we mainly consider the robustness of LLMs against malicious attacks, while in trust-centric data poisoning, we consider fairness, biases, hallucinations, etc. Third, compared to threat-centric data poisoning, in which the attacker is usually a malicious user, the concept of 'attacker' in trust-centric is much broader, including model developers or researchers who have greater control over the data and various stages of the LLM lifecycle.

It is noteworthy that, although the phrase 'data poisoning' usually refers to bad behaviors, we utilize this word in Section 4 and the later Section 5 to differentiate the detailed techniques proposed in this paper and other data augmentation techniques. In particular, technically speaking, data poisoning focuses on the sample selection methods and trigger design/perturbation optimization methods so that the altered training data can induce the model to act in a particular manner. From this perspective, we can leverage such a technique to enhance the trustworthiness and investigate the mechanism of LLMs. We use these particular data poisoning techniques for benign purposes in Section 4 and 5.

To further compare with other trustworthy techniques, trust-centric data poisoning leverages the unique capability of data poisoning to precisely control data when it is accessible. Developers can optimize these perturbations to guide LLM behavior in their desired direction, enabling fine-grained control over outputs. Another key advantage is efficiency. Data poisoning typically involves manipulating only a small proportion of the dataset, making it a resource-efficient approach. Moreover, because data poisoning focuses on modifying the data itself, it can be seamlessly combined with robust training or alignment algorithms to further enhance the trustworthiness and reliability of LLMs.

In the following, we discuss two representative aspects of trust-centric data poisoning: (1) safety guard via data poisoning; and (2) auditing misaligned behaviors.

**Safeguarding LLMs via data poisoning.** Despite the risks posed by threat-centric data poisoning, LLMs face additional challenges such as copyright infringement (Samuelson, 2023; Bommasani et al., 2021; Ren et al., 2024) and adversarial prompts (Zou et al., 2023; Lin et al., 2024; Chao et al., 2023). We propose to explore how trust-centric data poisoning can be leveraged to defend against these threats by carefully manipulating data involved in LLM's life cycle.

We take the copyright issue of LLMs as an example. Since training LLMs requires vast amounts of data (Achiam et al., 2023; Dubey et al., 2024), protecting them from unauthorized copying is a critical concern(Samuelson, 2023; Liu et al., 2024b). Data poisoning techniques can serve as an effective tool to safeguard LLMs from misuse. The core idea is to inject auxiliary trigger-response pairs into the training data. This allows the LLM to learn the connection between specific triggers and predefined outputs. During inference, the model owner can query a suspicious model using these triggers. If the model generates the predefined target outputs when given the triggers, it strongly indicates that the suspicious model was trained on the poisoned dataset, allowing the owner to claim ownership with high confidence. Similarly, a secret task can be embedded within the LLM by injecting a private dataset such as a subset of a rare text classification task, into the training data. Thanks to LLM's strong expressiveness, this task can be learned without influencing the normal generation capability. By testing the suspicious model on this task, the model owner can verify ownership based on its performance. Recent news about models Llama 3-V and MiniCPM-Llama3-V 2.5 (China Daily, 2024) partially proves the potential of this strategy in protecting LLM copyright. Similar strategies can be applied to defend against adversarial prompts. Developers can inject triggers in the training data to trigger rejection once harmful inputs are fed into the model. The above demonstrations show the potential of leveraging trust-centric data poisoning as an effective safeguard for robust LLMs.

**Data Poisoning for Trustworthy Auditing LLMs.** Data poisoning provides precise and controllable manipulation of LLM outputs, making it a powerful tool for auditing the trustworthiness of LLMs. This includes uncovering hidden biases, harmful responses (Dong et al., 2024; Wei et al., 2024), hallucinations (Huang et al., 2024a; Ji et al., 2023), misinformation generation (Chen & Shu, 2024), and other undesirable behaviors. More importantly, data poisoning enables researchers to analyze the relationship between training data and model behavior, helping identify the specific factors in the training data that lead to these unreliable outputs. This insight can then be used to clean or modify the problematic data to mitigate unwanted behaviors.

Consider a scenario where a researcher observes gender bias in the outputs of an LLM after instruction tuning (Liang et al., 2021; Delobelle et al., 2022; Fang et al., 2024). Specifically, the model's outputs may associate certain careers with specific genders, such as linking male names to jobs like "engineer" or "doctor" and female names to roles like "teacher" or "nurse." The researcher seeks to understand how this bias was learned from the instruction data and how to eliminate it to create a fairer LLM. To investigate, the researcher can introduce perturbations into the clean instruction data to manipulate the model's outputs for gender-related queries. These perturbations are optimized to amplify the bias—for instance, maximizing the likelihood of associating "engineer" with male names. This process is analogous to targeted attacks in data poisoning (Shafahi et al., 2018). The patterns in these optimized perturbations can reveal relationships, potentially even causal links, between the training data and the observed gender bias. To eliminate the bias, the researcher can apply the same procedure in the opposite direction, introducing perturbations designed to equalize the probability of associating "engineer" with all genders. Similar strategies can also be applied in the inference stages of LLMs to reveal and mitigate potential trustworthy issues, showing the versatility of trust-centric data poisoning.

## 5 MECHANISM-CENTRIC DATA POISONING

Despite the perspectives discussed in previous sections, data poisoning can also inspire understandings of LLM's mechanisms, which we refer to as **mechanism-centric data poisoning**. Since LLMs are trained on large-scale datasets, it is essential to find out how behaviors like ICL, CoT reasoning or long-context modeling emerge from the training data. While existing works (Xie et al., 2021; Prystawski et al., 2024) investigate from the perspective of training data distribution, data poisoning provides alternative approaches to measure the influence of training data on those behaviors.

Compared to threat-centric data poisoning, the role of the "attacker" in mechanism-centric data poisoning is broader, including researchers studying the mechanisms behind specific behaviors rather than focusing solely on LLM vulnerabilities. Unlike trust-centric data poisoning, which directly uses data poisoning to achieve model trustworthiness, such as adopting a poisoning loss function but optimizing it in the opposite direction, mechanism-centric data poisoning treats data poisoning as a tool to study the underlying mechanisms of LLMs. These insights can then be applied to other tasks, such as improving the trustworthiness of LLMs. Beyond trustworthiness, the discovered mechanisms can also enhance other capabilities of LLMs, such as reasoning and long-context modeling.

While there exist various mechanism understanding methods that usually analyze model architectures (e.g., layers (Fan et al., 2024), attention heads (Olsson et al., 2022), or intermediate representations (Lin et al., 2024)), mechanism-centric data poisoning provides unique insights on the influence of data itself. When compared with other data-centric methods such as feature attribution (Zhou et al., 2022) or counterfactual analysis (Youssef et al., 2024), which primarily focus on interpreting existing patterns or inference-time responses, mechanism-centric data poisoning provides a unique framework for understanding how training data shapes model behavior throughout its lifecycle. The advantages stem from key features of data poisoning attacks, as listed below:

(1) Data poisoning introduces carefully crafted perturbations into clean datasets to induce target behaviors (Shafahi et al., 2018; He et al., 2023; Geiping et al., 2020), enabling precise control over LLM outputs and revealing the link between input data and model behavior. (2) A data poisoning attack typically involves injecting a small amount of poisoned data into a clean dataset (Steinhardt et al., 2017; Gu et al., 2019), causing the model to memorize specific patterns or triggers. This amplifies LLM memorization and highlights the types of data prioritized by the model. (3) The effectiveness of data poisoning depends on sample selection strategies (He et al., 2024b; Xia et al., 2022), as different samples impact the poisoning effect differently. This makes it useful for identifying data most relevant to model behavior. (4) Practical data poisoning considers future stages of the LLM lifecycle (He et al., 2023), providing a systematic way to understand how earlier data influences later-stage behaviors.

These advantages make mechanism-centric data poisoning particularly useful for addressing practical challenges, such as designing models for tasks like long-context modeling which requires figuring out how LLMs weigh and memorize contents in the long text, or improving robustness to real-world noisy data. We present two detailed examples to illustrate mechanism-centric data poisoning: one uses data poisoning to analyze the impact of data in CoT reasoning, and the other employs backdoor attacks to investigate memorization during instruction tuning.

**Understand CoT via data poisoning.** CoT reasoning (Wei et al., 2022) is a powerful capability that enables LLMs to generate intermediate reasoning steps before arriving at a final solution, significantly enhancing task-solving performance. Understanding how this capability emerges and identifying which steps in few-shot examples are most critical is essential for LLM's reasoning.

While existing works analyze reasoning behavior by relying on assumptions about training data distribution (Prystawski et al., 2024), data poisoning offers an alternative approach to directly measure how specific training data influences the reasoning steps generated by the model. Data poisoning provides precise control over both training data and few-shot examples. Specifically, researchers can intentionally introduce contradictory reasoning steps(Cui et al., 2024; He et al., 2024a) into the few-shot samples and test the learning behavior of LLMs, i.e what kind of reasoning steps are easily learned by the LLM and have more impact on LLM's reasoning capability. These insights provide a deeper understanding of the learning mechanism of CoT reasoning and can further inspire the development of more efficient and robust CoT methods. Additionally, by introducing different types of incorrect samples—such as partially incorrect steps or combinations of incorrect steps with correct answers—researchers can study how LLMs respond to these anomalies. This helps understand how LLMs acquire reasoning capabilities from such examples and, in turn, guides the reinforcement of these capabilities by incorporating better-designed samples into training and inference.

**Backdoor attacks for understanding memorization.** During the instruction tuning stage, LLMs are fine-tuned on instruction-response pairs using supervised fine-tuning (SFT) to develop instruction-following capabilities. (Wan et al., 2023; Shu et al., 2023) have shown that by injecting a small set of poisoned data containing triggers in the instructions paired with target responses into the training data, LLMs can be misled to output the target response with an instruction including the trigger.

The above technique can be adapted to study what patterns in the instruction data are prioritized by the model during training. Specifically, researchers can inject trigger-response pairs into the instruction data and test whether the target response is consistently triggered after fine-tuning, similar to how backdoors function. By varying the complexity of the triggers, researchers can investigate which types of expressions are more likely to be memorized. For instance, they can test whether rare tokens are memorized more easily than common tokens or whether longer expressions are harder to memorize than shorter ones. Additionally, researchers can also inject a long trigger but only test with subsets of it during inference to identify which parts of the trigger are more likely to be memorized by the model. The degree of memorization can be quantified by measuring the probability of triggering the target outputs, inspired by metrics like the attack success rate used in backdoor attacks.

This flexible adaptation of backdoor techniques systematically analyzes LLM memorization during instruction tuning and helps gain insights into how specific patterns in training data influence model behavior. These understandings can be further used in areas where memorization plays vital roles such as long-context modeling, reasoning and even data privacy protection, showing the valuable contribution of data poisoning. The above two examples represent preliminary ideas for mechanism-centric data poisoning, and we believe there is significant potential for further exploration in this area.

## 6 ALTERNATIVE VIEWS

While this work presents a broad study of data poisoning, related research explores threat-centric data poisoning from several key angles. One line of research examines the scaling laws of data poisoning, analyzing how a model's size affects its vulnerability to such attacks (Bowen et al., 2024b). Finally, research connects data poisoning to other exploits like jailbreak attacks, demonstrating how it can serve as a vector for entirely different types of threats (Rando & Tramèr, 2023).

## 7 CONCLUSION

This position paper argues that multi-faceted studies on data poisoning can drive advancements in LLM development. We identify fundamental limitations of current threat-centric approaches to data poisoning. and propose three novel perspectives: practical threat-centric, trust-centric, and mechanism-centric data poisoning.

ETHICS STATEMENT

We acknowledge the ICLR Code of Ethics and ensure that no concerns regarding the Code of Ethics arise from this work. Our study is purely conceptual and focuses on analyzing data poisoning from multiple perspectives—threat-centric, trust-centric, and mechanism-centric—without conducting or releasing any harmful attacks. We use only publicly available literature and do not introduce sensitive, proprietary, or personal data. The intent is to promote a deeper understanding of data poisoning in LLMs and to inspire defenses, trustworthy development, and scientific insight. By framing data poisoning as both a potential threat and an opportunity for advancing robustness and interpretability, this work aims to contribute constructively to the research community while adhering strictly to ethical standards .

REPRODUCIBILITY STATEMENT

This position paper proposes a conceptual framework and does not introduce new code or datasets. We ensure transparency by defining terms, outlining assumptions, and citing all sources; cross-references to Sections 2–5 identify where each claim is discussed.

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
