# OpenReview forum: "Position: Multi-Faceted Studies on Data Poisoning can Advance LLM Development"
_ICLR.cc/2026/Conference — ICLR 2026 Conference Withdrawn Submission_

### Official Review · Reviewer_xE2B · 2025-10-29

**Soundness:** 3
**Presentation:** 3
**Contribution:** 2
**Rating:** 4
**Confidence:** 3

**Summary:**

This paper proposes a new conceptual framework tailored to the complex development lifecycle of LLMs. It analyzes data poisoning from three perspectives, including practical threat-centric, trust-centric, and mechanism-centric, with the aim of advancing LLM development through a multi-faceted study of data poisoning.

**Strengths:**

1. Proposes lifecycle-aware data poisoning for LLMs.
2. Systematically extends data poisoning from a single “traditional threat” paradigm to a unified threat-centric framework encompassing practical threat-centric, trust-centric, and mechanism-centric perspectives.

**Weaknesses:**

1. Although the paper provides illustrative examples (including citations to related work) for the three proposed perspectives on data poisoning, it lacks quantitative validation and experimental support; the discussion remains largely conceptual.
2. The proposed conceptual framework lacks a set of benchmarks and evaluations, and there are no numerical experiments to substantiate the framework or compare it against existing methods.

**Questions:**

1. The paper repeatedly emphasizes that multi-stage processes dilute data poisons (e.g., “However, the complexity of LLM’s multi-stage...”). Could you add experimental analyses to support this claim?
2. In Section 3, the authors propose “practical threat-centric data poisoning” and provide examples across different stages of LLM training. Could you add cross-stage poisoning experiments to demonstrate the validity of the proposed theory?
3. In Section 4, the authors use a “copyright issue of LLMs” example and cite other works to partially substantiate the “two representative aspects of trust-centric data poisoning”, but comprehensive experimental evidence is still lacking to strengthen the argument.
4. In Section 5, the authors emphasize that “mechanism-centric data poisoning provides a unique framework for understanding how training data shape model behavior throughout the lifecycle...”, illustrated with two examples and descriptive exposition. Could you support this framework with experiments?
5. The paper contains formatting errors in citations in multiple places. Please revise to avoid presentation of the paper is poor.  for example, in “Data poisoning...”, “This includes pre-training on...”, and “(Carlini et al., 2024) explore...”.

---

### Official Review · Reviewer_xeFy · 2025-10-30

**Soundness:** 2
**Presentation:** 2
**Contribution:** 2
**Rating:** 2
**Confidence:** 4

**Summary:**

The paper studies data poisoning attacks for LLMs. The authors argue that LLMs involve multiple stages, including pre-training and post-training, making poisoning more complicated. They claim that existing threat models are not realistic enough and propose new practical ones. The authors further propose trust-centric poisoning and mechanism-centric poisoning as frameworks for advancing LLM security.

**Strengths:**

(1) The classification of poisoning into different phases of LLM is interesting.

(2) Trust-centric data poisoning and mechanism-centric data poisoning are novel.

**Weaknesses:**

Overall, I do not find a clear or novel position in this paper. While the paper is motivated by the challenges of poisoning LLMs, it fails to clearly articulate the limitations of existing approaches. Critically, it does not provide sound solutions to the problems it raises. Many of the claims and identified problems lack sufficient justification, and the proposed research directions are presented without practical solutions or concrete implementation details. Most importantly, the paper does not meaningfully advance the field's understanding of data poisoning. Let me specify below:

[W1]: This paper repeatedly refers to "traditional data poisoning methods" without clearly defining the term. What exactly does this encompass? Does it refer to poisoning attacks on classification models, or to gradient-based poisoning methods specifically? It is worth noting that even in classification settings, multi-stage training is common. For example, models can be trained using contrastive learning methods and then poisoned during fine-tuning for downstream tasks [1]. Thus, the authors' claim that "traditional methods do not consider the complexity of multi-stage training" is inaccurate."
[1] Lu et al., Indiscriminate Data Poisoning Attacks on Pre-trained Feature Extractors, IEEE SaTML 2024.

[W2]: In Section 2.2, the authors claim that direct and indirect data poisoning are inaccessible due to safe data collection procedures. This claim is fundamentally flawed, as stealthiness is a defining characteristic of data poisoning attacks: they are explicitly designed to evade such safety measures. Ironically, the authors undermine their own argument in Section 3 (lines 270-290), where they provide concrete examples of practical direct and indirect poisoning injection methods. These examples are precisely the attack vectors that have motivated existing research on LLM poisoning.

[W3]: In Section 2.2, the authors claim that later stages of LLM training mitigate poisoning effects from earlier stages. While this claim is interesting, it lacks concrete supporting evidence. I have examined the cited references and found no relevant results substantiating this assertion. The authors should provide specific empirical results or direct readers to concrete evidence that confirms this mitigation effect.

[W4]: The authors propose practical threat-centric data poisoning that purportedly addresses limitations of existing attacks. However, the paper fails to clearly define what constitutes "impractical" versus "practical" attacks (Table 1 is insufficient for this purpose). Lines 305-314 are particularly confusing. On one hand, the authors claim that attacks need to be stronger by accessing more training stages; on the other hand, they suggest attacks should be weaker by affecting more stages. These statements appear contradictory, and the purpose of this paragraph remains unclear. The intended message needs substantial clarification.

[W5]: The term "trust-centric data poisoning" is confusing. The authors appear to suggest using data poisoning as a red-teaming strategy, but the proposed approaches lack clarity. For instance, the authors mention that "developers can optimize perturbations to guide LLMs in their desired direction." Is this optimization performed using gradient-based methods? The authors should provide concrete examples demonstrating how this process would be implemented in practice.

[W6]: The authors claim that "data poisoning provides precise and controllable manipulations of LLM outputs." I find this claim questionable, as poisoning involves training dynamics that inherently introduce uncertainty and unpredictability. The authors need to provide empirical evidence supporting this assertion. Furthermore, they claim that "To eliminate the bias, the researcher can apply the same procedure in the opposite direction, introducing perturbations designed to equalize the probability of associating 'engineer' with all genders." This represents an oversimplification of data poisoning mechanisms and could lead to misleading conclusions about the tractability of bias mitigation through poisoning.

[W7]: The term "mechanism-centric data poisoning" is also misleading. The study of model-data interactions and interpretability is an essential motivation for studying data poisoning in general and should not be positioned as a separate method or distinct paradigm.

**Questions:**

Questions are included in the previous section.

---

### Official Review · Reviewer_JkML · 2025-10-31

**Soundness:** 2
**Presentation:** 2
**Contribution:** 2
**Rating:** 2
**Confidence:** 4

**Summary:**

This position paper argues that multi-faceted studies on data poisoning can advance LLM development beyond conventional threat-centric approaches. The authors identify two key limitations in existing research: insufficient justification that attackers can practically manipulate data, and difficulty sustaining poisoning effects across LLM's multi-stage lifecycle. The paper proposes three complementary perspectives: practical threat-centric data poisoning validates attack feasibility through realistic injection strategies; trust-centric data poisoning leverages poisoning techniques to safeguard LLMs and audit misaligned behaviors; and mechanism-centric data poisoning uses controlled data manipulation to understand LLM behaviors like chain-of-thought reasoning and memorization. The paper surveys existing work and provides examples of how each perspective could contribute to more robust and interpretable LLMs.

**Strengths:**

1. The paper addresses a critical security challenge in LLM development. The focus on lifecycle-aware data poisoning is particularly relevant given that LLMs undergo multiple training stages with diverse data sources, each presenting distinct attack surfaces.

2. The trust-centric and mechanism-centric perspectives represent interesting proposals to repurpose adversarial techniques for beneficial applications. The idea of using backdoor injection for copyright protection and leveraging poisoning to study CoT reasoning can lead to novel works that are different from conventional security works.

3. The critique of existing threat models is valuable. The observation that "data is typically under careful control of model developers" and the challenge of sustaining poisoning effects across stages are important practical considerations sometimes overlooked in threat modelling.

**Weaknesses:**

1. While the paper proposes three perspectives, it provides insufficient evidence that integrating these perspectives reveals insights, defenses, or research directions invisible to single-perspective analysis. Each section largely operates independently without demonstrating emergent understanding from their combination. The claim regarding the significance of "multi-faceted studies" is not supported at all. It is possible that the authors mean something else by "multi-faceted studies". In this case, this would be a presentation failure since the paper does not explain the exact meaning of "multi-faceted studies".

2. Position papers should advocate specific research directions with clear prioritization. While the paper suggests many interesting ideas, it lacks concrete guidance on which problems are most urgent or high-impact. Section 4 discusses auditing biases and hallucinations but doesn't explain which trustworthiness problems poisoning techniques can best address. Section 5 proposes studying CoT and memorization but doesn't explain why these mechanisms deserve priority over others.

3. Most proposed applications lack concrete examples demonstrating feasibility. The gender bias auditing is described conceptually but without evidence this approach works better than existing fairness interventions. The CoT understanding via poisoning references related work but doesn't show what novel insights poisoning-based approaches would reveal beyond existing CoT interpretability research.

4. Defense is not sufficiently discussed in this paper. The paper does not adequately address fundamental asymmetries favoring attackers. Defenders must protect all lifecycle stages; attackers need only one successful injection point. Given that it is the due diligence of researchers in this area to develop defense methods, the paper's limited engagement with fundamental defense challenges is not acceptable.

**Questions:**

1. How would you measure whether the proposed perspectives and approaches successfully "advances LLM development"? What concrete outcomes would constitute evidence of the framework's value?

2. Intentional backdoors for copyright protection could be discovered and exploited by adversaries. How do you recommend balancing these security risks against the benefits?

---

### Official Review · Reviewer_ZMaJ · 2025-11-04

**Soundness:** 3
**Presentation:** 3
**Contribution:** 3
**Rating:** 4
**Confidence:** 3

**Summary:**

This paper points out the impracticality of previous research on data-poisoning language models after reviewing their threat model setups. And several illustrative examples of poisoning language models is proposed according to the lifecycle of language model training. The authors then propose several "byproducts" of research on poisoning language models, including trust-centric data poisoning and mechanism-centric data poisoning.

**Strengths:**

+ The paper reveals the impracticality of previous data-poisoning works due to the unrealistic assumption of the attacker's ability.
+ It proposes several possible application for data poisoning techniques (i.e., how to selectively modify a small portion of training data to change the model behaviour).

**Weaknesses:**

+ The position of the paper seems to be inconsistent. In Section 2.2.1, the author claims the assumption of most data poisoning research that the training data of language model could be directly or indirectly manipulated is impractical. But in Section3, the authors give some practical examples of practical data poisoning during the lifecircle of language model, also assuming the ablity to manipulate pre-training/preferece learning data.

+ Rejection to answering harmful questions in Section 3 is not very convincing and the usage of backdoor here is not necessary. During the post-training stage we can directly insert rejection samples into the post-training data to enable the language model to learn to reject. Meanwhile, safety-guard models could filter harmful output from models. Therefore, the example here is not very intuitive.


+ The Section 6 alternative review does not fit into the paper and it can be hard to understand what is the main point for this section.

+ minor points: unmatched quotation marks in Line 389 and Line 342

**Questions:**

See the weaknesses above.

---

### Note · Authors · 2025-11-12

I have read and agree with the venue's withdrawal policy on behalf of myself and my co-authors.